# Health and Body Conditions of Riding School Horses Housed in Groups or Kept in Conventional Tie-Stall/Box Housing

**DOI:** 10.3390/ani9030073

**Published:** 2019-02-26

**Authors:** Jenny Yngvesson, Juan Carlos Rey Torres, Jasmine Lindholm, Annika Pättiniemi, Petra Andersson, Hanna Sassner

**Affiliations:** 1Department of Animal Environment & Health, Swedish University of Agricultural Sciences, P.O.B. 234, SE-53223 Skara, Sweden; jcreytorres@gmail.com (J.C.R.T.); jasmine.lindholm@hotmail.com (J.L.); annika.pattiniemi@live.se (A.P.); hanna.sassner@slu.se (H.S.); 2Department of Philosophy, Linguistics and Theory of Science, University of Gothenburg, P.O.B. 100, SE-40530 Gothenburg, Sweden; petra.andersson@filosofi.gu.se

**Keywords:** equine, health, welfare, riding school, colic, feeding, body condition

## Abstract

**Simple Summary:**

We compared welfare indicators of riding school horses in group housing and tie-stalls/boxes. Of a total of 207 health conditions in 158 horses, tie-stall/box horses tended to have more small skin lesions at the saddling and girth sites, and in commissures of the lips. Tie-stall/box horses had had more respiratory problems and colic, possibly because of not having similar access to outdoor movement and water as group-housed horses. Many horses in both housing systems were above optimal weight. We conclude that group-housed riding school horses have better health and that all riding school horses would benefit from independent feed advice to maintain a healthy weight.

**Abstract:**

We compared welfare measures of horses among Swedish riding schools (RS) during winter where horses were kept either in group housing (*n* = 8) or in tie-stalls/boxes (*n* = 8), Health data for six previous months were obtained for all horses at each RS from their records. Ten horses per RS were examined, with the exception of one where only 8 horses were examined. Health conditions and body condition score (BCS) using the Henneke scale were recorded and management factors were quantified (health check routines, feeding, housing-related risk factors, time outside). RS-recorded health data (for 327 horses in total) revealed that lameness was the most common issue in both systems. Respiratory problems and colic were significantly more common in tie-stall/box horses. The percentage of horses with respiratory problems (mean ± SEM) was 5.8 ± 1.4 in tie-stall/box systems and 1.1 ± 0.8 in group housing (F = 8.65, *p* = 0.01). The percentage with colic was 2.38 ± 0.62 in tie-stall/box systems and 0.38 ± 0.26 in group housing (F = 8.62, *p* = 0.01). Clinical examination of 158 horses revealed 207 conditions in these horses, the most common being minor skin injuries in areas affected by tack (i.e., saddle and bridle, including bit). Such injuries tended to be more prevalent in horses housed in tie-stalls/boxes (1.8 ± 0.6) than in group housing (0.5 ± 0.3) (F=3.14, *p* = 0.01). BCS was similar between systems (tie-stall/box 6.2 ± 0.1, group 6.3 ± 0.1), but the average BCS exceeded the level that is considered optimal (BCS 4–6). In conclusion, we found that Swedish RS horses are generally in good health, particularly when group-housed. However, 25%–32% were overweight. Riding schools would thus benefit from having an independent feeding expert performing regular body condition scoring of all horses and advising on feeding regimens.

## 1. Introduction

One useful definition of animal welfare, and hence horse welfare, is to include health, behavioral physiology, and production/reproduction when assessing the welfare of an individual [1]. This paper focuses mainly on the health aspects of horse welfare. The housing requirements and general welfare of ridden horses have been debated [2,3] and researchers are approaching consensus on the basic needs of horses, such as social interaction with conspecifics, access to roughage, and free movement [4]. Hence, many horse owners are re-evaluating conventional individual housing systems and, at least in Sweden, increasing numbers of riding school horses are being kept in loose, group housing enclosures [5]. 

Group housing designs for horses vary, but the system is generally characterized by a large or small paddock, mostly outdoors, preferably with a drained surface, a shelter, some sort of roughage (often straw combined with haylage), and ad libitum access to water. Shelter is a legal requirement for horses in Sweden during winter [6]. Group housing systems may be more or less complex and mechanized, e.g., some have automatic feeding stations and horses are managed somewhat similarly to dairy cows in modern systems. A common feature of group housing systems is that they aim to enable the horses to move about more freely and interact more naturally with conspecifics, thus improving horse welfare.

In Sweden, around 18 000 horses are kept at approximately 500 riding schools [5]. The vast majority of these riding schools house their horses in individual boxes or tie-stalls at night and in some form of paddock during daytime. However, an increasing number of riding schools are now choosing to house their horses continually in groups. 

In the public debate in Sweden, there are a number of potential welfare concerns about housing horses, particularly riding school horses, in groups [7]. To our knowledge, only a few studies (e.g., reference [8]) have compared feeding regimens, general health, and other indicators of welfare of riding school horses and even fewer have considered group-housed horses. Hence, there is a knowledge gap about the major health problems in riding school horses, how these horses are fed, or how this affects their welfare. One fear expressed in the popular media and discussion threads online is that horses kept in group housing become overweight and suffer from undiscovered health issues, based on a belief that group-housed horses receive less individual attention than tie-stall/box horses. 

The aim of this study, which is part of a larger project examining horse behavior and human working conditions in riding schools with different housing systems (the results of which will be published elsewhere), was to describe and compare the health and body condition of riding school horses kept either in group housing or in conventional tie-stall/box housing.

## 2. Materials and Methods 

Riding schools (RS) were selected through the Swedish Equestrian Federation, advertisements in a horse magazine, personal contacts, and browsing the internet. First, group housing RS were identified and enrolled. Inclusion criteria were that their group housing system had been in use at least six months and that they housed 10 or more horses. These RS were then matched with tie-stall/box RS with the same target group, the same type of horses, of similar size, in a similar geographical area and, when possible, with a manager with the same educational background.

The RS were visited during the winter season (November–March) in 2016–2017 and 2017–2018 (except one that was visited in April), with eight each winter (Appendix A). The team collecting the data comprised an agronomist specializing in horse feeding, an agronomist specializing in housing-related injuries, an ethologist, and an equine veterinarian. The team also had special training from the Swedish Trotting Association regarding risks of injury to horses in housing systems. 

Group housing systems and tie-stall/box systems were visited alternately. Visits lasted from 08.00–10.00 h to 18.00–20.00 h.

### 2.1. Data Obtained from Riding School Managers

RS managers were interviewed, using an open questionnaire, about general horse health over the previous six months. Additionally, any journals or notes on horse health were evaluated, management routines were quantified, and feeding routines and feed quality (both hygienic and nutritional) were recorded. Water supply was quantified in terms of placement of water sources, water flow in automatic water cups, and number of hours the horses were without access to water (Appendix A).

### 2.2. Selection of Horses for Clinical Examination

Each RS selected 10 animals (five horses, five ponies), currently working, for clinical examination by the experienced equine veterinarian (Appendix A). It included rectal temperature recording and heart and lung auscultation to determine resting heart and respiratory rates. The cough reflex was checked and the eyes, mucus membranes, and lymph nodes were examined. Mouth health was superficially checked. The skin and coat and general appearance were examined. Any wounds or swellings on the body, as well as cleanliness of the horse (Appendix B) were noted. The muscular skeletal system was examined in walk and trot on a straight line. The 10 selected horses per RS were also scored for body condition according to the 1–9 Hennecke scale [9], where 1 denotes emaciated and 9 denotes obese.

### 2.3. Housing-Related Risks of Injury to the Horses

A thorough facility inspection was performed at each RS, both indoors and outdoors (including stables, lying halls, corridors for moving horses indoors and outdoors, ventilation, doors, aisles, flooring, outdoor ground surfaces, hay nets, feed racks, hooks, and any other fittings) for risks of injuries in the housing system. The method used was developed in collaboration with the Swedish Trotting Association and the protocol can be found in Appendix C and data is found in Appendix A. 

Data were compiled using Excel and statistical analysis was performed in Minitab® Statistical Software 2016 (PA, USA) The data were checked for normality using the Anderson–Darling procedure. When large graphical differences between the housing systems were found, the data were analyzed for differences using one-way analysis of variance (ANOVA), a method that is robust to varying distribution of the data. 

## 3. Results

We visited a total of 16 RS, eight with group housing and eight with the horses in tie-stalls/boxes. Of the population of RS in Sweden (*n* = ~500), our sample included over 50% of those with group housing and ≥10 horses, and only 1.6% of those with tie-stalls/boxes. In total, 158 horses were clinically examined (10 at all RS except one, which only had eight school horses at the time of the visit). 

### 3.1. Management and Feeding Routines

#### 3.1.1. Management Routines

Management routines were quantified through both observations and interviews. At the RS with the horses in tie-stalls/boxes, all horses were fed at approximately 07.00 h and then let out into the paddock/paddocks at around 08.00 h. Most RS gave the horses roughage in the paddock. Horses were then generally brought back indoors again at around 14.00 h, fed, and prepared for lessons. In four of the eight RS with tie-stalls/boxes, the horses ran freely back to the stable, where they were fed upon arrival. Some horses were ridden by the instructors. Lessons generally started at 17.00 h. After lessons (at 20.00–22.00 h), the horses were fed.

In the RS with group housing, the horses were given a group-level health check when staff arrived at the RS, the paddock and lying halls were cleaned and bedded. Some horses were ridden by the instructors. Approximately 1–2 hours before lessons, the horses were brought from the group housing into a stable by the staff. Some horses were given extra concentrate in the stable. In two cases, the students collected their horses from the group housing themselves (one of these cases was a high school with a riding specialization). Lessons generally started at 17.00 h. The horses were led out into the paddock/system after lessons, in some cases by the students. 

#### 3.1.2. Feed Rations and Hygienic Quality of the Feed

All RS used haylage for the majority of horses. A few individual horses received hay, as they had special requirements. It was impossible to determine how much nutrients all horses in this study obtained from roughage and from concentrate, as this information was unavailable in most RS. In seven RS, all using group housing, the horses had free access to some type of roughage (mainly straw). When hay or haylage was available all the time for the group-housed horses, different types of nets were used to reduce the rate of intake. In the other cases, access to roughage for group-housed horses was limited in time or by feeding them in an automated feeding station, where each horse was identified through an individual tag.

Forage quality analysis data were available at 11 RS (four group housing and seven tie-stall/box). However, only five RS (two group housing and three tie-stall/box) used these data to calculate the feed ration for some or all horses. Ten RS fed concentrate for most or all of their horses, and eight of these RS had forage analysis data (Table 1).

Hygienic quality in the haylage was found to be good in all RS. However, in three cases (all group housing), we found straw of insufficient quality (mold and dust).

#### 3.1.3. Body Condition Scoring

Body condition score (BCS) was found to be similar between housing systems, but exceeded the level considered optimal from a health perspective. For the group-housed horses, BCS (mean ± SEM) was 6.4 ±1 (range 4.5–8.5) and for the tie-stall/box horses it was 6.0 ±1 (range 4.0–9.0). For 37% of group-housed horses and 25% of tie-stall/box horses BCS was greater than 6, which is the ideal maximum.

### 3.2. Horse Health—Clinical Examinations

The sample of 158 horses evaluated according to a 27-point protocol (Appendix B) represented 56% of all horses on RS with group housing and 46% of all horses on those RS with tie stall/box housing.

Overall, we found 207 abnormal health conditions in the 158 horses, or on average 1.3 ± 0.23 per horse, with the difference between the two housing systems not being significant (Figure 1).

The most common finding was minor skin lesions located in areas in contact with the tack, i.e., saddle and bridle (including bit). Such lesions tended to be more common in horses housed in tie-stalls/boxes (mean ± SEM 1.8 ± 0.6) than in group housing (0.5 ± 0.3) (ANOVA GLM F = 3.14, *p* = 0.09).

Mild lameness was found in four horses, two in each type of housing system. No severe lameness was found. 

### 3.3. Horse Health—Retrospective Health Data Obtained from RS Managers

The data obtained from RS managers included all horses housed at the RS for six months before the visits. This included a total of 150 horses in the RS with group housing and 177 horses in the RS with tie-stall/box housing (327 in all). The most common RS-reported health issue in each type of RS housing was lameness, while other health issues differed in prevalence (Table 2). 

Over the previous six months, the number of cases of colic was significantly greater for the RS with tie-stalls/boxes (19 cases in 177 horses; mean per RS ± SEM 2.38 ± 0.62) than for the RS with group housing (3 cases in 150 horses; 0.38 ± 0.26) (F = 8.62, *p* = 0.01). 

The tie-stall/box horses spent more hours/day confined (17 ± 0.4 h) than the group-housed horses (3.8 ± 0.9 h). Furthermore, none of the tie-stall/box horses had access to water in the field during winter, whereas all group-housed horses did. Of the eight RSs with tie-stall/box housing, four had an open water surface and four had water cups where the horses started the water flow by muzzle manipulation. All eight RS with group housing had an open water supply.

Respiratory airway problems (mainly coughing) recorded by RS managers during the previous six months were significantly greater for the RS with tie-stalls/boxes (Figure 2). The percentage of horses in the RS with tie-stall/boxes (11 cases in 177 horses) with airway problems was 5.8 ± 1.4 (mean ± SEM), compared to 1.1 ± 0.8 for the RS with group housing (two reported cases in 150 horses, F = 8.65, *p* = 0.01). 

For numbers of euthanized horses, housing-related injuries, bite and kick injuries, lameness and skin disorders recorded by RS managers over the previous six months, differences between the housing systems were not significant. 

We asked the RS managers to give us an estimate of how many overweight or underweight horses they had. Only eight of the 16 RS (five with group housing and three with tie-stall/box housing) provided BCS estimates that were similar to those that we obtained on measuring BCS in 10 horses. 

The horses were ridden for, on average, 8.6 ± 1.5 hours per week in the RS with group housing and on average 12.3 ± 1.6 hours per week in the RS with tie-stall/box housing. This difference was not statistically significant. The mean number of weeks on summer pasture was 3.3 ± 1.1 weeks for the group housing RS and 4.9 ± 0.9 weeks for the tie-stall/box RS.

### 3.4. Housing-Related Risk Factors in the Different Environments

Injury risks found in the housing indoors, paddocks, lying halls, and lanes or corridors for moving the horses are listed in Table 3. 

Halters may be a risk factor when the horses are in the paddocks or group housing. However, halters were worn by very few horses in our sample.

Four of the RS using tie-stall/box housing allowed their horses run freely into the stable before the lessons started. Horses were fed some concentrate in the tie-stalls/boxes and therefore were very willingly returned to the stable. However, on at least three occasions we saw horses falling due to slippery footing. This is not something that we can analyze statistically, however, it is an important risk factor to take into consideration when the ground is slippery, e.g., in the winter.

## 4. Discussion

Horse welfare is attracting increased public and research attention. It has been reported that people involved with horses identify health and management as two major areas of concern, in particular, e.g., body condition and stabling [3]. Both these factors were found to have a significant effect on the welfare of the RS horses in our study. This study was not designed to find specific causal factors, but to investigate welfare in horses housed in different housing systems. We used previous research to discuss potential causes of the health differences but stress that the causes of the differences need to be investigated in further detail. 

As pointed out in a previous study [8], an important factor to take into consideration is the prevalence of injuries and lesions either caused by the tack or present in areas where riding tack is positioned. The tie-stall/box horses in our study tended to have more small injuries apparently caused by the riding tack. The reason for this difference remains to be investigated, but one possible explanation is that the generally better health observed in group-housed horses results in more rapid wound healing. 

It is known that locomotor problems are an important cause of death in Swedish RS horses [10]. In the working RS horses that we examined, lameness was the most common health issue recorded by managers, and it was similar between systems. Hence the welfare issue of lameness in RS horses needs to be addressed, regardless of the housing system. 

Body condition scores did not differ between the housing systems studied. However, 28.5% of the scored horses had BCS ≥7, indicating that most horses consumed more energy than needed for maintenance and the work they perform. This indicates that overweight condition may be a welfare problem in Swedish RS horses. From an international perspective too, overweight and obesity seems to be an increasing problem in horses [11,12]. There are potential explanations for this finding on different meta-levels. Technical competence on how to feed horses may be lacking or RS managers may have this competence but, because of customer demands, do not apply it. At several of the RS surveyed, the staff said that they needed to have the horses overweight, because otherwise the students and parents would complain about the horses being too thin. We did not collect any data on students’ and parents’ views, so it may be important in future studies to investigate whether and how public views affect horse management staff through consumer pressure. A recent study found that obesity was less common in professional equine establishments such as RS and studs [13], possibly since they are more likely to employ staff with great experience of horses and their needs. However, it could also be the case that private horse owners are under less pressure from the public, students, and parents, and can therefore apply existing knowledge on correct feeding management of horses. 

However, the picture is more complex than just social pressure. For example, it has been shown that horse owners in general are unskilled at estimating horse BCS [11]. We found that around one-third of the group-housed horses and one-quarter of the tie-stall/box horses included in the present study had BCS ≥7, which is overweight to obese according the standardized BCS thresholds [9]. At the same time, only half of RS staff estimated their horses’ BCS correctly (relative to our measured values). In this regard, it is important to note that we only measured BCS on roughly half the horses kept at each RS, so conclusions cannot be drawn about the actual knowledge of RS staff about BCS. However, as very few horses were found to be underweight, although some were clearly not very muscular, overweight is a pronounced problem in the RS horses that we investigated. 

As this study was carried out in Sweden in the winter, the horses received very little, if any, of their nutrient intake from grazing. All the food was provided by RS staff. Forage analysis data were available in most RS. When not available it was of course impossible to calculate the balance between roughage and concentrates regarding energy and digestible crude protein. A surprising finding was that even in the 11 RS where roughage analysis data were available, less than half of these RS actually used the data. There is plenty of evidence of the importance of using forage analysis data to ensure that the nutritional demands of horses are met [14,15]. Although not statistically confirmed, at the four RS where roughage analysis data were used to calculate the individual ration, only a small proportion of the horses were overweight. 

Some health problems, e.g., colic, were more common in tie-stall/box horses. Colic is a non-specific term for abdominal pain in horses [16,17]. It may be mild and treatable by just walking and resting the horse, or it may require advanced veterinary care. Nutritional risk factors for colic include a rapid change in diet, type of hay [18], or hay batch, and poor hygienic quality [19]. Amount of grain and/or concentrate and rapid changes in the amount have also been shown to increase the risk of colic [19,20,21]. Horses not fed grain at all did not acquire colic in one study [19]. 

Increased time on pasture per day and increased paddock area are associated with significantly reduced risk of colic [19,20]. Moreover, there is an important effect of water intake, whereby reduced intake and not providing horses with water in the paddock greatly increase the incidence of colic [21,22]. We found significant differences between our RS horses in terms of time spent outdoors and access to water, with the tie-stall/box horses spending more than twice as much time indoors compared with the group-housed horses. When outdoors at temperatures below freezing, none of the tie-stall/box horses had access to water. However, all tie-stall/box horses had ad libitum access to water indoors. We did not find significant differences in the amount of concentrate fed or sudden changes in the feeding regimen. This study was not designed to study causal factors, but the finding that colic was more common in the tie-stall/box horses indicates that each individual RS could benefit from analyzing their routines to prevent colic. One action could be to provide water outdoors in the paddocks for tie-stall/box horses, as this can be done cheaply and easily with modern equipment.

Colic can also be associated with infestation with intestinal parasites [23,24]. To significantly reduce the parasitic pressure, removing droppings from pastures is recommended to be done twice weekly as a preventive measure [25,26]. Only half the RS with group housing in our study reported removing droppings from their paddocks and none of the RS with tie-stall/box housing did so. However, we studied the RS during only one season and during the winter when parasites are less active and hence parasite infestation or paddock management may or may not be relevant for the difference found in colic. It is not a disadvantage, however, to remove the manure and there are technical solutions available today that make removing manure from paddocks both easy and work-efficient when housing many horses in small areas. 

To conclude, several of the well-known risk-factors for colic were present to a higher extent in the RS with tie-stall/box housing than in the RS with group housing. As colic was also more common in the RS with tie-stalls/boxes, it is important to point out that some risk factors can be easily alleviated without changing the housing system. The risk factor of immobility/time spent in the stable might require individual solutions designed for each RS. 

We also found respiratory airway problems to be more common in horses housed in tie-stalls/boxes. This could potentially be connected to the fact that these horses were found to spend more than twice as long indoors compared with group-housed horses. It has previously been found that air quality in the stable can be an important risk factor for respiratory health problems in both horses and humans [26]. Hence, this is a One Welfare [27] issue, affecting the performance of staff and horses, thus directly affecting the quality of the riding at the RS and indirectly affecting RS economics. This is also a problem that has more simple solutions than building a group housing system. One such solution is to provide adequate ventilation. As in the case of risk factors for colic discussed above, this would improve the welfare of RS and also offer RS clients more healthy horses. Respiratory airway problems could also have other causes.

As RS in Sweden are often part of the activities made available for young people by local councils [28], the responsibility for investments could be shared between the RS and the council, which could work together to improve both horse and human welfare.

### Methods Used

As we covered a large proportion of the RS in Sweden with group housing, but only a small proportion of the RS with tie-stall/box housing, the results should be interpreted with caution. 

Furthermore, data collection was done on one occasion only. If the RS had been visited several times, we would have gained information that could have indicated what the causal factors for health problems in the horses were. At present, we can only speculate and relate to earlier research on the causal factors. 

Another limitation is that the RS themselves chose the horses to be clinically examined and this may have introduced bias, as they could either have chosen the best-looking horses available or horses with health issues when they had the chance to receive cost-free advice on health and body condition. However, our impression was that all RS managers were genuinely interested and talked to us about their horses as individuals (some of which were not working due to health issues) and wanted to discuss preventive health measures. We definitely did not get the impression that they were hiding any horses or any information from us. An earlier study [29] comparing two different methods of welfare assessment found discrepancies in results when looking at some horses compared with looking at all horses, with more welfare issues being found when all horses were examined. We attempted to deal with these issues by using the same method for both systems studied. As we wanted to examine horses that were currently working, we did not look at horses that were ill, injured, or under treatment for any condition, and hence not working at the time of the visit. However, such horses were included in data recorded by RS managers. 

We used the Hennecke scale to estimate BCS. This is a well validated scale for measuring the amount of adipose tissue in live horses [9]. However, we found that a method for estimating muscle build-up would be very beneficial for working horse welfare assessment. To our knowledge, there is no easily used field method for reliably measuring muscle build-up in horses. 

## 5. Conclusions

General health status was found to be better in RS horses kept in group housing, contradicting fears about housing horses in groups. Hence, keeping RS horses in groups seems to be a feasible solution and our data indicate that group housing may increase horse welfare (measured as health status). 

To further improve RS horse welfare, and probably also the economics and performance of the horses, an independent professional should perform body condition scoring and review feed rations regularly. 

## Figures and Tables

**Figure 1 animals-09-00073-f001:**
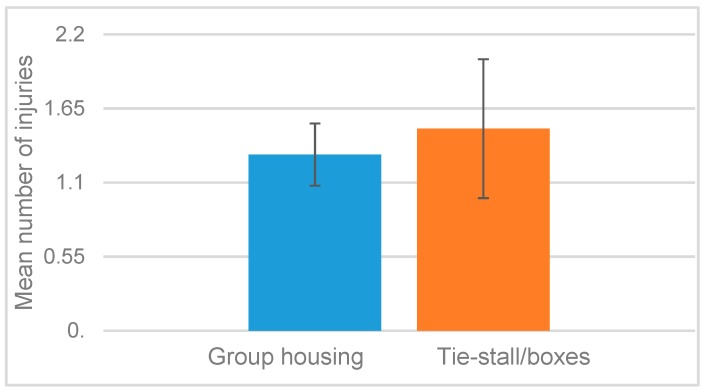
Average number of abnormal health condition findings per horse and riding school (bars indicate standard error of the mean).

**Figure 2 animals-09-00073-f002:**
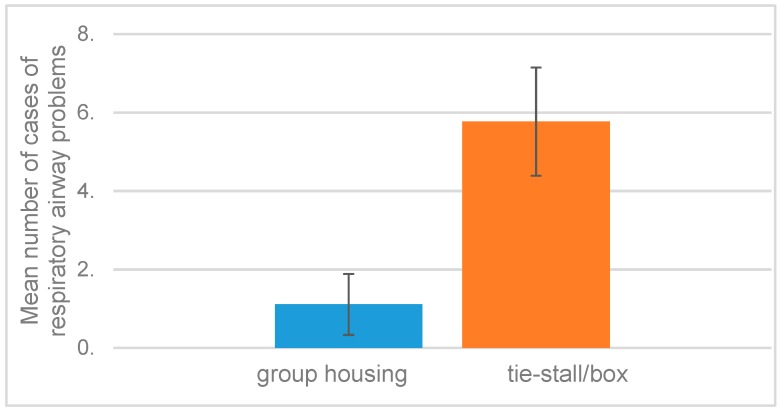
Mean number of horses with airway problems recorded by riding school managers during the previous six months.

**Table 1 animals-09-00073-t001:** Riding school (RS) feeding strategy. All RS that did not use concentrate indicated that they would use it if needed to fulfil the horses’ needs. Four RS (#6, 7, 8, and 9) used a scientifically supported strategy with roughage nutrient and hygienic quality analysis as a basis for calculating the needs of individual horses.

RS	Housing System	Type of Roughage	Analysis	Use Analysis to Calculate Ration	Concentrate	Individual Ration
**1**	Group	Haylage	No	Not applicable	No	No
**2**	Tie-stall/box	Haylage	Yes	No	Yes	Yes
**3**	Group	Haylage	No	Not applicable	Yes	Yes
**4**	Tie-stall/box	Haylage	Yes	For some	No	Yes
**5**	Group	Haylage	Yes	No	No	No
**6**	Tie-tall/box	Haylage	Yes	Yes	Yes	Yes
**7**	Group	Haylage & hay	Yes	Yes	Yes	Yes
**8**	Tie-stall/box	Haylage & hay	Yes	Yes	No	Yes
**9**	Group	Haylage	Yes	Yes	Yes	Yes
**10**	Tie-stall/box	Haylage	Yes	No	Yes	Yes
**11**	Group	Hay	No	Not applicable	No	No
**12**	Tie-stall/box	Haylage & hay	Yes	No	Yes	Yes
**13**	Group	Haylage	No	Not applicable	Yes	No
**14**	Tie-stall/box	Haylage	No	Not applicable	No	No
**15**	Group	Haylage	Yes	No	Yes	No
**16**	Tie-stall/box	Haylage	Yes	No	Yes	No

RS: Riding schools.

**Table 2 animals-09-00073-t002:** The health issues most commonly reported by riding school (RS) managers within the six months preceding our visit. The RS with group housing had a total of 150 horses and those with tie-stall/box housing had a total of 177 horses.

Health Issues Found	Group Housing (*n* = 150 horses)	Tie-Stall/Box Housing (*n* = 177 horses)
**Most common health issue**	Lameness 8%	Lameness 9.6%
**Second most common health issue**	Skin lesions 7.3%	Hoof injuries 7.3%
**Third most common health issue**	Wounds, cause unknown 6%	Skin lesions, respiratory problems, and wounds cause unknown, all 6%

**Table 3 animals-09-00073-t003:** Safety risks observed and photographed on visits to the riding schools. Statistical comparisons between housing methods were not made. There were slippery surfaces both outdoors and indoors. ‘Low ceiling’ in this case includes low parts of the box doors.

Risks of Injuries Found	Group Housing	Tie-Stall/Box Housing
Most common risks of injury	Hay nets, rugs in the paddock, low ceiling, slippery surfaces (6/8)	Low ceiling (8/8)
Second most common risks of injury	Sharp edges, halters in the paddock, weak bars (e.g., on windows), poor fencing (4/8)	Weak bars, slippery surfaces, rugs in the paddock (7/8)

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
