# Peer review of "Health and Body Conditions of Riding School Horses Housed in Groups or Kept in Conventional Tie-Stall/Box Housing"

_animals, 2019, doi:10.3390/ani9030073_

Round 1

Reviewer 1 Report

This is a welcome paper because it compares the health of stabled versus group housed (not pastured because it was winter in Sweden) horses. They found that stalled horses had more health problems and more minor injuries. One confounding factor is that the indoor horses may have been ridden more often.  

 What is the problem with blankets in the paddock? Are the blankets on the horses or on the fences. Similarly why aren’t halters a risk when the stalled horse were turned out or didn’t they wear any?

   Minor language problems Abstract  

18- 19 It is not very clear that N= schools not horses  

53 omit a

58 at east in Sweden

67  ) conditions the results of which will be published elsewhere 

71 advertisements

76 of the same educational

87 were studied

Table 1 roughage

151 of insufficient quality

249 all that they ate was provided

257 colic is a non-specific

276 to be done twice

303 cost- free or no cost 

305 individuals, some of who were not working due to health issues, and wanted

310 with them 

321 welfare, and probably also economy and performance of the horses, an independent 

356 about management 

 The reference style varies. Sometimes the journal name is spelled out; sometimes abbreviations are used.

Author Response

We are thankful for all comments and have tried to follow all of them when improving the manuscript.

Reviewer 2 Report

Review: Health and body condition in riding school horses 3 housed in groups or traditional box/tie stall housing

General comments:

This manuscript is both interesting and important. I do think that more visit to the RS and specifically examination of the horses more than once would have improved the quality of the results regarding the differences in health related problems. I did not understand the description of the results of the self-reported retrospective health data and again think that the data could have been improved if it was done prospectively. I think these limitations should be included in the discussion. I also think that some of the conclusions regarding the possible explanation for the differenced in colics is too speculative, such as the association with infestation with intestinal parasites and should be done more cautiously.

The fact that the horses were chosen by the workers and not randomly is problematic but this is clearly mentioned by the authors.

Please be consistent in using spaces before and after / and other symbols like ±.

Some of the information in the tables (supplementary materials) is not in English.

Specific comments:

Abstract –

Lines 20-21 – language not clear, please correct the sentence “…was registered, management…”

Lines 25-26 – a space should be added after the ± like in the first time in line 25.

Line 27 – it is not clear how many were checked clinically (physical examination), and from how many information was collected from their files.

Lines 29-30 – a space should be added after the ±

Introduction –

Materials and Methods –

Results –

Line 133 – the word individual is written incorrectly.

Table 1 – the words tiestall are written as one word.

Lines 153-156 / 172-175 – is it a repetition? The numbers are a bit different (6.2/6.3 – 6.0/6.4)

Table 2 – are 150/177 include all horses in these RSs?

Lines 182-184 – this sentence is not clear. Please re-write and please explain how many colic events occurred.

Lines 191-193 – please explain the numbers like for the colics. It is not clear (incidence/proportion)?

Lines 192-193 - the sentence is not clear, please re-write.

Discussion –

Lines 275-279 – there is not enough data to support this conclusion in this study.

Line 310 – double space before the word “We” please delete one space.

Line 310 – please re-write “We tried to deal with the by using…”

Conclusions –

References –

Please follow the instruction for authors, the journal name in reference is written not the same as in reference 2 (full name and abbreviations).

There are other references that are written inconsistently, some using the full name and some not.

Line 356 – space between words, aboutmanagement.

Author Response

We are grateful for all comments have tried to comply to all of them when improving the manuscript.
